# Resilience of the Skin Microbiome in Atopic Dermatitis During Short-Term Topical Treatment

**DOI:** 10.3390/ijms262311737

**Published:** 2025-12-04

**Authors:** Malin Glindvad Ahlström, Rie Dybboe Bjerre, Yue Hu, Maike Seifert, Fredrik Boulund, Lone Skov, Jeanne Duus Johansen, Lars Engstrand

**Affiliations:** 1National Allergy Research Centre, Department of Dermatology and Allergy, Herlev and Gentofte Hospital, University of Copenhagen, 2900 Hellerup, Denmark; riedybboeolsen@gmail.com (R.D.B.); jeanne.duus.johansen@regionh.dk (J.D.J.); 2Department of Dermatology and Allergy, Herlev and Gentofte Hospital, University of Copenhagen, 2900 Hellerup, Denmark; lone.skov.02@regionh.dk; 3Centre for Translational Microbiome Research (CTMR), Department of Microbiology, Tumor and Cell Biology, 171 77 Stockholm, Swedenmaike.seifert@ki.se (M.S.); fredrik.boulund@ki.se (F.B.); lars.engstrand@ki.se (L.E.); 4Department of Clinical Medicine, Faculty of Health and Medical Sciences, University of Copenhagen, 2200 Copenhagen, Denmark

**Keywords:** atopic dermatitis, dysbiosis, farnesol, microbiome, moisturiser, shotgun metagenomics, staphylococcus aureus, topical

## Abstract

Atopic dermatitis (AD) is associated with microbial dysbiosis and impaired skin barrier function. Topical therapies, such as moisturisers and antimicrobial fragrance compounds, may modulate the skin microbiome and support disease management. The objective was to evaluate how a moisturiser and a fragrance compound (farnesol) influence skin microbiome composition in individuals with AD and healthy controls. In a randomised, controlled, operator-blinded study, 15 AD patients and 15 healthy controls applied a moisturiser, farnesol, moisturiser + farnesol, or no treatment to defined skin areas over 7 days. Microbiome composition, alpha/beta diversity, and core taxa were analysed using shotgun metagenomics. At baseline, AD patients exhibited distinct microbial profiles, including elevated *Staphylococcus aureus* and *Micrococcus luteus*. Neither moisturiser nor farnesol significantly altered richness, beta diversity, or core taxa in either AD patients or controls. However, moisturiser use in healthy individuals modestly increased Shannon diversity, reflecting improved microbial evenness. Despite clear microbiome differences between AD and healthy skin, short-term topical treatment did not markedly shift microbial composition. The observed stability underscores the resilience of the skin microbiome and suggests that longer interventions or more targeted formulations may be necessary to influence microbial dysbiosis in AD.

## 1. Introduction

Atopic dermatitis (AD) is a chronic inflammatory skin disorder characterised by impaired skin barrier function, dryness, and pruritus. An important aspect of its pathogenesis is disruption of the skin microbiome, characterised in patients with AD by reduced microbial diversity and an overrepresentation of pathogenic organisms, particularly *Staphylococcus aureus* (*S. aureus*) [1]. These microbial shifts are believed to both contribute to and exacerbate disease severity [2,3,4].

Targeting the skin microbiome dysbiosis in AD by topical treatments such as moisturisers and fragrance compounds with antimicrobial properties has shown promising results in previous studies [5]. Moisturisers are widely used in the management of AD due to their barrier-restoring effects, and emerging evidence suggests that they may also enhance microbial diversity and richness, potentially supporting the re-establishment of a balanced skin flora [5,6]. Further, certain fragrance components have demonstrated selective antimicrobial action [7], which may help reduce pathogenic overgrowth, especially in AD-affected skin.

However, the precise impact of topical treatments on both normal and AD skin microbiomes remains unclear. Particularly, the effects on bacterial phylogenetic relatedness, alpha and beta diversity, and the abundance of specific bacterial lineages have not been comprehensively evaluated in controlled clinical settings.

The present randomised controlled study investigates how a moisturiser and a fragrance compound, both with antimicrobial activity, influence the skin microbiome in both healthy individuals and those with AD. We used Doublebase Gel™ as the moisturiser, a formulation previously used in large-scale eczema prevention studies, and farnesol as the fragrance, selected for its known antimicrobial activity against *S. aureus* [7,8,9]. We hypothesise that both moisturiser and fragrance treatment modulate the eczematous skin microbiome through antimicrobial effects, enhancing microbial diversity and potentially reducing *S. aureus* dominance in AD skin.

This study aims to provide deeper insight into how commonly used topical agents modulate microbial ecology and how such modulation might contribute to skin health and the management of AD.

## 2. Results

Cutaneous microbiome profiling was completed for 22 of 30 enrolled participants (13 atopic dermatitis patients and 9 healthy controls) (Appendix A). Eight microbial swabs were collected per participant (four at baseline and four at day 8). The study population characteristics are summarised in Table 1. In the AD cohort, the median EASI was 6.2 (IQR: 3.9–19.8), corresponding to mild disease. Overall, there were no clinical changes in the AD group over time (*p* = 0.06). The pre-defined treatment areas included both lesional and non-lesional skin. Local eczema scores decreased slightly from baseline to study completion in the moisturiser-treated area (mean change 0.87, *p* = 0.01), with no significant changes in the other treatment or control areas.

Baseline skin pH was higher in AD patients than in controls, although this difference did not reach statistical significance (5.63 vs. 5.43; *p* = 0.18). After moisturiser application, healthy controls exhibited a slight increase in skin pH (mean ΔpH = +0.22; *p* = 0.05), whereas the increase observed in the AD moisturiser subgroup was not significant (mean ΔpH = +0.17; *p* = 0.155). No pH changes were observed following fragrance application or in the combined fragrance and moisturiser groups. 

An overview of the bacterial community composition for all obtained samples for the two groups fulfilling the criteria of being prevalent in more than 30% of samples and detected at a minimum of 0.1% abundance (core species) is shown in Figure 1. Overall, our data demonstrated that the microbiome differed significantly between patients with AD and healthy controls (*p* < 0.05) (Figure 1 and Figure 2), suggesting distinct microbial communities between the two groups. Overall, the AD patients had more *Micrococcales* (*p* = 0.008), *Micrococcae* (*p* = 0.006), *Micrococcus* (*p* = 0.013), *Micrococcus luteus* (*p* = 0.013), and *S. aureus* (*p* = 0.002), and healthy controls had more *Actinomycetales* (*p* = 0.034) and *Kytococcus* (*p* = 0.027). The relative abundance of core taxa in pre-treatment samples showed that patients with AD had more *Micrococcus aloeverae* (*p* = 0.042), *Micrococcus luteus* (*p* = 0.027), *S. aureus* (*p* = 0.001), *S. capitis* (*p* = 0.037), and *S. epidermidis* (*p* = 0.000009), and these differences remained after treatment for *Micrococus luteus*, *S. aureus*, and *S. epidermidis*. Further, individuals had a distinct skin microbiome despite disease group (*p* < 0.001) (Figure 2B).

The beta diversity analyses found no change in the microbial communities by any treatment for all participants or within the AD or healthy control group (Figure 3B). In patients with AD compared to healthy individuals, no change in richness (number of species per sample) was found for any treatments. However, a significant difference in alpha diversity (estimated using Shannon’s diversity index) was observed in the moisturiser-treated control group when comparing pre- versus post-treatment (*p* = 0.023) (Figure 3A). No other groups showed significant differences in alpha diversity. There appeared to be no difference in the core species abundance.

In all samples (pre- and post), significant differences in the core species were observed between the AD and the control group (Figure 4). In the AD group, 18 core species were identified, while 17 core species were identified in the CT group. Of these, 15 core species were shared between both groups. In the AD group, three distinct species were present—*Kocuria rhizophilia*, *human endogenous retrovirus K*, and *S. aureus*—whereas two other species—*Corynebacterium lipophiloflavum* and *Streptococcus oralis*—were more present in control samples.

Significantly higher abundances of *M. luteus* (pre-treatment: *p* = 0.027; post-treatment: *p* = 0.00048), *S. epidermidis* (pre-treatment: *p* = 0.000009; post-treatment: *p* = 0.00048), and *S. aureus* (pre-treatment: *p* = 0.001; post-treatment: *p* = 0.001) were observed in AD patients before and after treatment. In contrast, *M. aloeverae* (pre-treatment: *p* = 0.042) and *S. capitis* (pre-treatment: *p* = 0.037) were elevated only in pre-treatment samples.

The relative abundance of the core species after each treatment, as tested by the Wilcoxon paired test, did not change after any treatment in the AD or control group (*p* > 0.05). ANCOVA analyses were conducted to compare the effect of each treatment (M, F, or MF) with the untreated control area (C) in the AD and the control cohort. In each ANCOVA, each patient’s pre-treatment sample was used as the baseline, and the effects from the three compared treatments were evaluated. A tendency toward changes in certain species was observed following treatment; however, after correction for multiple testing, none of these changes reached statistical significance (Appendix A).

## 3. Discussion

This randomised controlled study provides insights into how commonly used topical agents—a moisturiser and an antimicrobial fragrance—affect the skin microbiome in individuals with atopic dermatitis (AD) and healthy controls. Our findings reaffirm the significant differences in microbial composition between AD and healthy skin, with higher relative abundances of *S. aureus* and *M. luteus* in AD patients. These species, particularly *S. aureus*, have been consistently implicated in AD pathogenesis due to their pro-inflammatory potential and ability to impair skin barrier function.

Despite these baseline differences, neither the moisturiser nor the fragrance compound (farnesol), alone or in combination, produced significant alterations in overall microbial richness, beta diversity, or core species composition in either cohort. This suggests a relative stability of the skin microbiome over a short treatment period, even in the presence of agents with known antimicrobial and barrier-restoring properties. However, in healthy controls, moisturiser application led to a modest but statistically significant increase in Shannon diversity, reflecting improved microbial evenness. This finding may reflect a beneficial effect of moisturisers in promoting a more balanced skin microbial ecosystem, as has been shown by others [5]. In line with this, a shift toward a healthier stratum corneum lipid composition in atopic dermatitis skin was observed following moisturiser treatment in our patient cohort, published elsewhere [10]. Further, there was a slight clinical local eczema improvement in the moisturiser group of AD patients. Moisturisers are commonly used both as a standard treatment for eczema and by individuals without skin conditions. Although short-term moisturisers may not drastically shift microbial communities, they may subtly influence microbial balance, which may be relevant for managing AD. The composition of the moisturiser—specifically its ingredients—appears to play a significant role in shaping these microbial changes [6,11,12]. We used a moisturiser identical to that employed in a large-scale infant study investigating its potential prophylactic effect against eczema development [8]. It is plausible that alternative formulations, containing different ingredients, could exert distinct effects on the skin microbiome.

Fragrance ingredients are frequently used in moisturisers. Some plant-derived fragrance ingredients are known to have antimicrobial actions, for example, geraniol and farnesol. Despite prior evidence suggesting farnesol can selectively inhibit *S. aureus* [7,9], we observed no significant microbiome shifts following fragrance treatment. This may reflect the relatively low baseline abundance of *S. aureus* in our mildly affected AD cohort or limited antimicrobial potency at the concentration used. Farnesol’s impact may also depend on its formulation, anatomical site of application, and interaction with the host lipid environment—factors not fully explored in this study.

Our results corroborate and extend current knowledge of microbial dysbiosis in AD. Patients had non-significantly higher baseline pH compared with healthy controls, reflecting known barrier impairment in AD [13,14]. Since skin pH influences microbial ecology—favouring pathogenic bacteria like *S. aureus* in more alkaline environments—this shift may contribute to dysbiosis. Previous studies report *S. aureus* colonisation in up to 70 percent of lesional AD skin versus 39 percent of non-lesional skin, correlating with disease severity [2,3]. In the present study, in addition to *S. aureus*, *M. luteus* and other members of the Micrococcaceae family were increased in the AD group. Although Micrococcus species are common commensals, *M. luteus* has been reported at higher levels in AD skin and can enhance *S. aureus* proliferation and virulence [15,16]. Conversely, taxa such as Actinomycetales and the genus *Kytococcus* were more prevalent in healthy controls, consistent with their roles in maintaining stable skin microbiota [17].

The identification of overlapping yet distinct core species between AD and healthy skin further highlights the ecological divergence of skin microbial communities. While 15 core taxa were shared, *Kocuria rhizophila* and *S. aureus* were uniquely abundant in AD, whereas *Corynebacterium lipophiloflavum* and *Streptococcus oralis* were more prevalent in controls. Recent work in children with AD has shown that higher relative abundance of *K. rhizophila* is associated with lower disease severity and improved skin barrier function and that *K. rhizophila* can inhibit *Staphylococcus aureus* growth in vitro, suggesting a potential protective role in the atopic skin microbiome [18]. The presence of human endogenous retrovirus (HERV)-K in the AD group warrants further investigation into potential host-microbe interactions or sequencing artefacts. HERV, including HERV-K, has been implicated in inflammatory and neoplastic skin diseases, particularly psoriasis and melanoma. Altered expression of other HERV families, such as HERV-E, has also been reported in atopic dermatitis skin, but to our knowledge, HERV-K expression has not been systematically characterised in lesional atopic dermatitis [19].

Interestingly, although microbial communities in this study differed significantly between AD and control groups overall (*p* < 0.05), intra-individual microbiomes were highly distinct and stable, with individual variation being a dominant factor (*p* < 0.001). Multiple studies have demonstrated that the human skin microbiome exhibits a highly individualised composition, with each person harbouring a unique microbial signature that remains relatively stable over time. This individuality often exceeds differences observed between disease states [20,21]. Interindividual variability in the microbiome and its responses may partly explain why certain topical agents show variable clinical effects in practice, and therapeutic modulation may need to be tailored accordingly.

In a longitudinal study of 12 healthy adults monitored for up to 2 years, the microbiome remained stable and exhibited remarkable resilience to routine external exposures such as showering, climate fluctuations, and interpersonal contact [20]. Consistent with these findings, microbiome shifts induced by cosmetic use or cleansing regimens are often transient and reversible after product discontinuation [11,22].

### Strengths and Limitations

The strength of this study is the randomised controlled setup, taking the topographical [15,17,23], temporal [21], and inter-individual variation [24,25] in skin microbiome into account. The limitations include the limited sample size and mild disease of the patients with AD. Further, the intervention period was short, and conclusions on long-term applications cannot be elucidated. In addition, only one concentration of fragrance was used.

## 4. Methods and Materials

### 4.1. Study Population

An experimental clinical randomised and operator-blinded case–control study was conducted during 10 months in 2020/2021. Eligible participants were Caucasian and 18 years or older. Fifteen had a diagnosis of AD established according to the UK Working Party Diagnostic Criteria [26], with disease onset in childhood and, at the time of the study, mild to moderate eczema characterised by regular flare-ups and visual eczema within the preceding three months. An additional fifteen age- and sex-matched healthy controls were included. Exclusion criteria for all participants included pregnancy, breastfeeding, fragrance allergy, and active infection; healthy controls were additionally excluded if they had a history of eczema. Participants with AD were excluded if they had concomitant dermatological disease or had severe eczema at the test sites.

Recruitment followed approval by the ethics committee of the Capital Region of Denmark (H-18058392) and the Danish Data Protection Agency (P-2020-717).

### 4.2. Restrictions

During and four weeks prior to entering the study, UV treatment, use of topical and/or systemic antibiotics, probiotics, and fungicides were prohibited. Further, subjects were instructed not to use chlorinated pools and saunas and to avoid fragrance and topical treatment on their arms within 7 days of enrolment until the last study day. Fragrance-free soaps and shampoos were provided to all participants to be used during the study period.

### 4.3. Study Design

The study was conducted at the Department of Dermatology and Allergy at Herlev and Gentofte Hospital over a one-week period and included two study visits. At baseline, blood samples were taken for filaggrin (FLG) genotyping, and a clinical assessment was performed using the Eczema Area and Severity Index (EASI). Four predefined skin areas (5 cm × 10 cm) on the upper inner arm and volar forearm were randomised (right/left allocation) to receive either no treatment (C), moisturiser (M), fragrance (F), or a combination of fragrance and moisturiser (MF). Participants applied each treatment three times daily for seven days, following a standardised protocol that included hand washing prior to application and the use of separate gloves for each arm.

At baseline and at day 8, at least 12 hours after the last application, the four test areas were clinically scored for eczema severity (local eczema score), pH was measured, and the areas were microbially swabbed. The local eczema score ranged from 0 to 15 and was based on severity ratings [1,2,3] for erythema, oedema, excoriation, lichenification, and dryness. Further, tape strips were collected from all sites to analyse stratum corneum lipids; these results are reported elsewhere [10].

The moisturiser was 0.2 mL Doublebase Gel ™ (Dermal Laboratories, Herts, UK) containing 15% *w*/*w* isopropyl myristate, 15% *w*/*w* liquid paraffin, glycerol, carbomer, sorbitan laurate, trolamine, phenoxyethanol, and purified water. The fragrance was farnesol (3,7,11-Trimethyldodeca-2.6.10-Trienol (CAS [4602-84-0]) in aqueous solution at a concentration of 1000 ppm (0.1%), with 50 µL applied per treatment. Farnesol was mixed with Doublebase Gel for the MF applications.

Skin pH was measured with the Mettler-Toledo Seven2Go pH metre (mV metre S2 with a surface probe) (Sigma-Aldrich^®^, St. Louis, MO, USA). The mean value of triplet measurements was used.

From blood samples, genomic DNA was purified and typed for the FLG loss-of-function mutations R501X, 2282del4, and R2447X [27].

### 4.4. Microbial Skin Swab Collection and Processing

For each test area, a moist swab was rubbed across the entire area for 30 s, excluding the distal regions reserved for pH measurements. The swab was then rotated in a PowerBead tube containing CD1 solution, after which the swab tip was cut off and stored in a freezer.

### 4.5. Extraction of Skin DNA and Shotgun Metagenomic Sequencing

Each sample swab and the associated buffer were thawed and transferred to Bashing Bead lysis tubes (Zymo Research, Irvine, CA, USA) containing a mixture of 0.1 mm and 0.5 mm-sized beads. To validate the assay, three positive controls (75 µL ZymoBIOMICS Microbial Community Standard in 725 µL DNA/RNA Shield, Zymo Research) and three negative controls (800 µL of DNA/RNA Shield, Zymo Research) were included in the extraction.

Mechanical lysis was performed using a FastPrep24 5G bead beater (MP Biomedicals, Santa Ana, CA, USA). Samples were subjected to five cycles of bead beating at 6.0 m/s for 1 min, with 5 min intervals between cycles, resulting in a total bead beating duration of 5 min. Following lysis, samples were centrifuged at 10,000× *g* for 1 min. A 200 µL aliquot of the resulting supernatant was then transferred to a deep-well plate and purified using the ZymoBIOMICS 96 MagBead DNA Kit (Zymo Research) on a Tecan Fluent automated liquid handling platform (Tecan, Männedorf, Switzerland). DNA was eluted in 50 µL of EB buffer (Qiagen, Hilden, Germany) and stored at −20 °C until further analysis.

Shotgun metagenomic libraries were prepared with the MGI FS DNA library preparation kit (MGI, Shenzhen, China) according to the manufacturer’s instructions and sequenced on a DNBSEQ-G400 sequencer (MGI, Shenzhen, China) using a DNBSEQ-G400RS high-throughput sequencing set, generating 150 bp paired-end reads (FCL PE150, MGI, Shenzhen, China).

### 4.6. Sample Selection and Bioinformatics Analysis

Sample selection followed the flowchart outlined in Appendix A. Experimental controls and sequencing controls were excluded from the downstream analyses after inspecting their read counts and microbial community profiles.

High-throughput sequencing data were analysed using the StaG-mwc (v.0.5.0 https://zenodo.org/records/8032462, accessed on 9 November 2025) metagenomic analyses pipeline [28]. For each sample, 20.5 million reads were randomly selected for subsequent analyses. Initial reads preprocessing was performed using Fastp (v.0.23.0) with default settings as defined in StaG-mwc. Host reads were removed by mapping to the masked human genome (GRCh38) using Kraken2 (v2.1.2), as implemented in StaG-mwc. Taxonomic profiling was performed with MetaPhlAn3 (v.3.0.14) against the mpa_v30_CHOCOPhlAn_201901 MetaPhlAn database with additional command-line flags “--unknown_estimation” and “--add_viruses” within the StaG-mwc pipeline.

### 4.7. Statistical Analysis

All statistical analyses were performed using R (v.4.1.2). Alpha-diversity (Chao1 and Shannon indices) was assessed, and pairwise comparisons were conducted using the Wilcoxon test, with Benjamini–Hochberg correction for multiple comparisons. Bray–Curtis distances were calculated to quantify inter-sample dissimilarity, and PERMANOVA (1000 permutations) was employed to evaluate group differences. To adjust for baseline differences between individuals, ANCOVA was performed to determine post-treatment changes in species. Prior to ANCOVA, data were centred log-ratio (CLR) transformed. Robust linear regression was used to fit the ANCOVA models. Results were converted to log2 fold changes for comparisons. Differences in pH between patients and controls were assessed using unpaired *t*-tests, while longitudinal changes in pH, as well as EASI and local eczema scores within each treatment group, were assessed using paired *t*-tests.

## 5. Conclusions

Our data underline the persistence of core dysbiosis signatures in AD despite topical intervention, emphasising the complexity of microbiome modulation in chronic inflammatory conditions. The absence of significant treatment effects may reflect the relatively short application period or the mild disease severity among participants. Moisturiser use in healthy individuals modestly increased microbial evenness, indicating a potential role in promoting microbial balance. These results suggest that effective microbiome modulation in AD may require longer treatment durations, more targeted formulations, or adjunctive approaches to better explore the therapeutic potential of microbiome-based interventions.

## Figures and Tables

**Figure 1 ijms-26-11737-f001:**
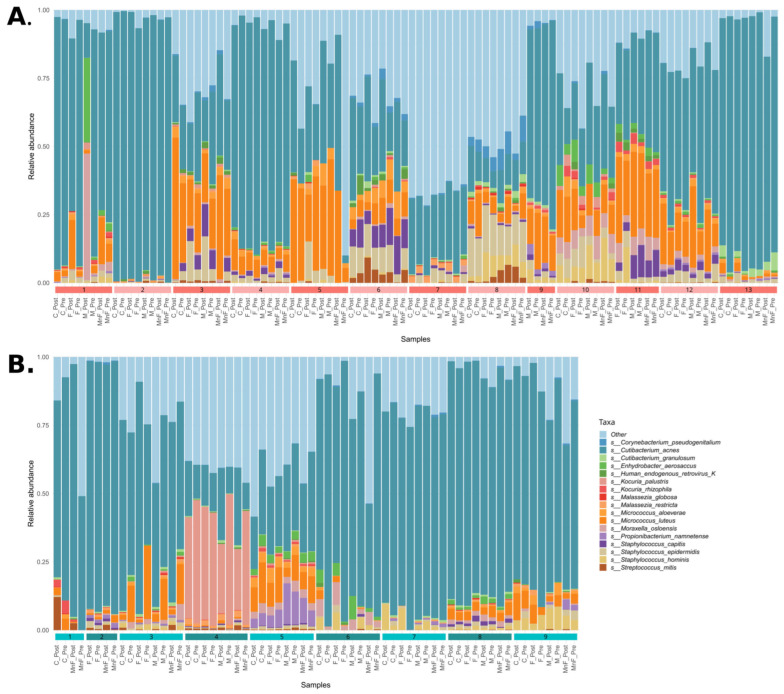
Bar plots of the community composition of species that were (1) present in more than 25% of samples and (2) with a relative abundance > 0.1% in the samples. (**A**) Patients with atopic dermatitis and panel (**B**) healthy controls. Samples are grouped by individual, indicated by coloured numbered bars along the bottom of each bar plot.

**Figure 2 ijms-26-11737-f002:**
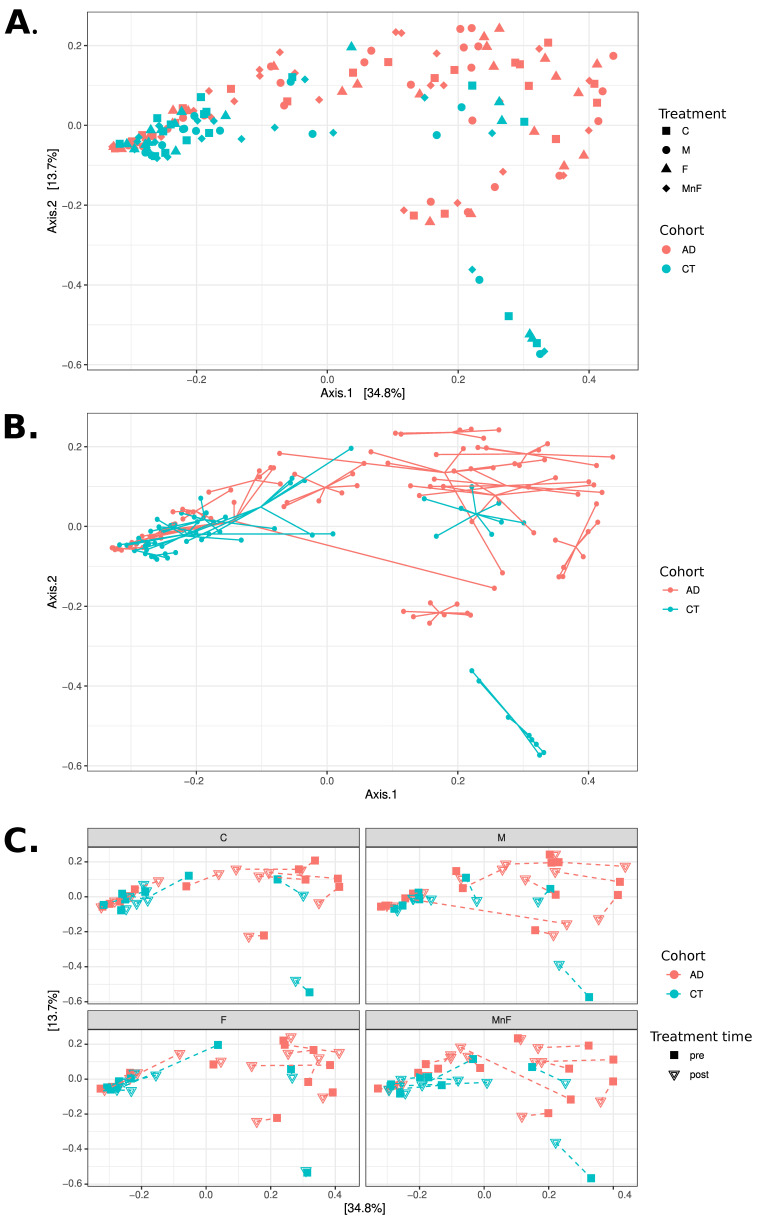
PCoA analyses at the species level using Bray–Curtis distances. All three panels display PCoA plots of the same samples in the same coordinate system. (**A**) Samples coloured by cohort, with shape indicating treatment (C, M, F, MnF). Note how the CT cohort appears mostly concentrated to the left, while the AD cohort is mostly located in the upper right of the figure. (**B**) Samples coloured by cohort, with samples belonging to the same individual connected with lines. Note how individuals appear to have mostly distinct personal microbiome compositions. (**C**) Four subgraphs showing each treatment group in separate subplots using the same coordinate system, coloured by cohort. The shape of points indicates treatment time (pre vs. post), with each individual’s pre- and post-samples connected with a dashed line. Note that only a few individuals exhibit substantial changes in composition between pre- and post-treatment.

**Figure 3 ijms-26-11737-f003:**
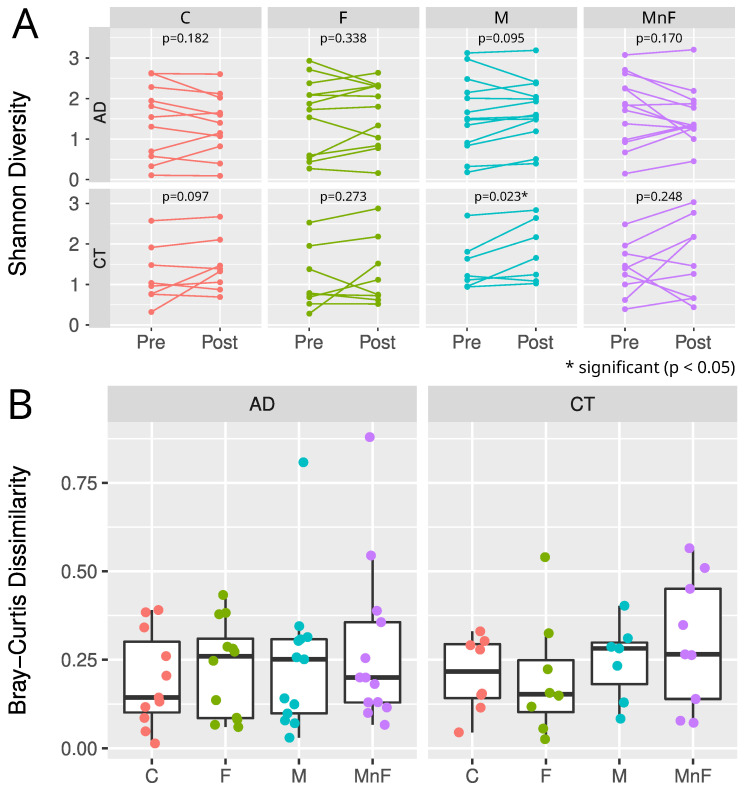
(**A**) Alpha diversity estimated by Shannon diversity for each cohort, treatment, and time point. Wilcoxon paired test was used to estimate *p*-values for pre- vs. post-differences for each cohort and treatment. Note how the control cohort exhibited statistical differences (*p* = 0.023) in pre- vs. post-moisturiser treatment. (**B**) Boxplots showing median, upper, and lower quartiles of Bray–Curtis distances of pre- and post-samples, grouped by cohort and treatment. Each sample is shown by coloured dots. Note that there are no statistical differences in the microbiome composition in any treatment group, despite the two potential outliers in M and MnF treatments in the AD group.

**Figure 4 ijms-26-11737-f004:**
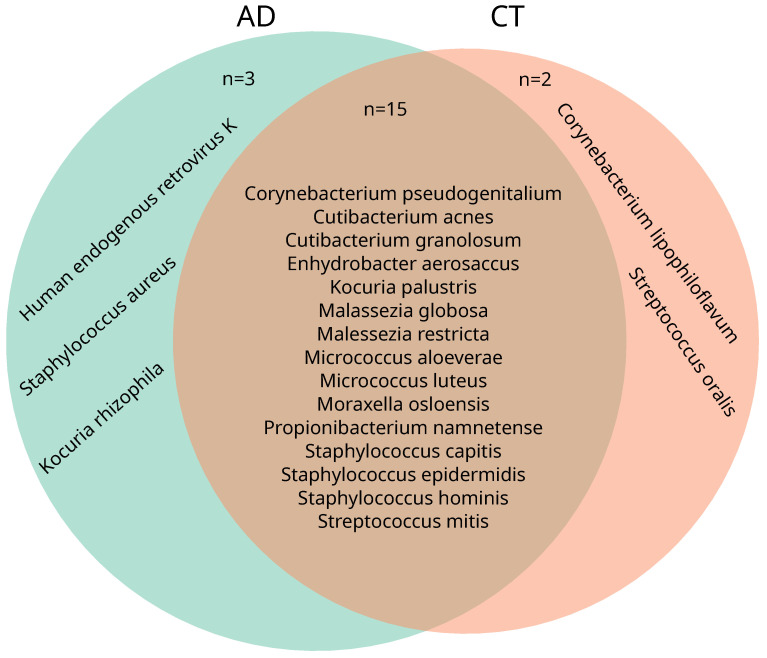
Venn diagram of core species from the two cohorts. Core species: at least 0.1% relative abundance found in at least 30% of samples from the sample group. Both cohorts show an overlap in most identified core species (15 overlapping species), with the exception of three species unique to the AD cohort: *Kocuria rhizophilia*, human endogenous retrovirus K, and *Staphylococcus aureus*, and two species unique to the CT cohort: *Corynebacterium lipophiloflavum* and *Streptococcus oralis*.

**Table 1 ijms-26-11737-t001:** Clinical characteristics of the two study groups, values in number (%) if not otherwise noted.

Characteristic	AD Patients (*n* = 13)	Healthy Controls (*n* = 9)
Age years (median (IQR))	25 (23–58)	25 (23–48)
Sex, female, *n* (%)	7 (53)	6 (67)
FLG-mutation (%)	1 (7.7)	1 (11.1)
EASI (median (IQR))	6.2 (3.9–19.8)	-

## Data Availability

The datasets presented in this article are not readily available because of ethical restrictions. Requests to access the datasets should be directed to the corresponding author.

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
