# Peer review of "Resilience of the Skin Microbiome in Atopic Dermatitis During Short-Term Topical Treatment"

_ijms, 2025, doi:10.3390/ijms262311737_

Round 1
Reviewer 1 Report
Comments and Suggestions for Authors
The manuscript describing the impact of a clinical intervention (skin moisturizer, a fragrance, or both) on atopic dermatitis' skin microbiota is very well written and designed. The comparison with healthy controls allowed for a basal comparison and additional information. Figures are adequate, and the text is concise and clear. I have just minor questions to present.
- Why is alpha diversity not presented in a figure? Shannon's index may be presented as a bar graph as beta diversity is shown. Figure 3 may include both diversity analyses.
- The last phrase of page 4 says that "...a significant difference in alpha diversity (estimated using Shannon’s diversity index) was observed in the control group when comparing pre versus post treatment (p=0.023)." Which of the three treatments was responsible?
- The description of the skin sites that were sampled does not include the notion of the disease. Was it lesional skin in all AD cases and sites? Or was it unaffected skin from AD patients? Or a mix of both lesional and unaffected skin? Please clarify this point in the Materials and Methods section.
Author Response
- Why is alpha diversity not presented in a figure? Shannon's index may be presented as a bar graph as beta diversity is shown. Figure 3 may include both diversity analyses.
Answer: Thank you for your great suggestion. Alpha diversity estimated by Shannon diversity for each cohort has now been included in figure 3 as suggested.
- The last phrase of page 4 says that "...a significant difference in alpha diversity (estimated using Shannon’s diversity index) was observed in the control group when comparing pre versus post treatment (p=0.023)." Which of the three treatments was responsible?
Answer: Thank you for this clarifying question. It is in the moisturizer group and has been added to the manuscript.
- The description of the skin sites that were sampled does not include the notion of the disease. Was it lesional skin in all AD cases and sites? Or was it unaffected skin from AD patients? Or a mix of both lesional and unaffected skin? Please clarify this point in the Materials and Methods section.
Answer: Thank you for this helpful comment. As described in the methods section, the sampling areas were pre-defined and randomized, resulting in a mix of both lesional and non-lesional skin in participants with AD. We have now clarified this explicitly in the Materials and Methods section.
Reviewer 2 Report
Comments and Suggestions for Authors
In their manuscript entitled “Resilience of the skin microbiome in atopic dermatitis during short-term topical treatment” Ahlström et al. analyzed the composition of the skin microbiome in AD patients vs healthy controls after application of a moisturizer and a fragrance. They could confirm changes in the microbiome in AD patients compared to healthy controls, but besides subtle changes no difference after application of the moisturizer and fragrance or a combination thereof. The manuscript is well written and the study is well controlled, however some questions remains to be answered.
Introduction part:
Please mention already in the introduction which moisturizer and which fragrance was choosen and why?
Study design: The authors analyzed EASI scores before treatment, was there a change in EASI by the treatment?
Since farnesol is a contact sensitizer, was there any effect of farnesol on skin appearance or EASI score?
The authors observed the abundance of Staph aureus, Kocuria rhizophila and human endogenous retrovirus K in AD patients. What is known for the latter 2 species in AD?
Figure 1A especially is rather busy. Can there be a summary of all 13 AD patients and may be also for Fig 1B for the control patients?
Author Response
- Reviewer 2:
In their manuscript entitled “Resilience of the skin microbiome in atopic dermatitis during short-term topical treatment” Ahlström et al. analyzed the composition of the skin microbiome in AD patients vs healthy controls after application of a moisturizer and a fragrance. They could confirm changes in the microbiome in AD patients compared to healthy controls, but besides subtle changes no difference after application of the moisturizer and fragrance or a combination thereof. The manuscript is well written and the study is well controlled, however some questions remains to be answered.
Introduction part:
Please mention already in the introduction which moisturizer and which fragrance was choosen and why?
Answer: Thank you for this helpful comment. We have now added a description of the specific moisturizer and fragrance used, including the rationale for their selection, in the Introduction. The moisturizer (Doublebase Gel™) was chosen because it has been used in a large-scale infant study assessing its potential prophylactic effect against eczema, making it relevant for studies of skin barrier modulation. Farnesol was selected as the fragrance due to its documented antimicrobial properties, including selective activity against S. aureus. These details have now been incorporated into the revised manuscript.
Study design: The authors analyzed EASI scores before treatment, was there a change in EASI by the treatment?
Answer: Thank you for this relevant question. As described in the first part of the Results section, we did not observe any overall change in EASI scores following treatment. The local eczema score showed a slight decrease in the moisturizer-treated area, whereas no change was observed in the areas treated with fragrance or the combined formulation.
Since farnesol is a contact sensitizer, was there any effect of farnesol on skin appearance or EASI score?
Answer: Thank you for this important point. We did not observe any visible skin reactions including eczema in response to farnesol. The concentration used in the present study (0.1%) was substantially lower than that used for patch testing (4% pet.) and in line with cosmetic industry guidelines — which likely explains the absence of visible skin reactions.
The authors observed the abundance of Staph aureus, Kocuria rhizophila and human endogenous retrovirus K in AD patients. What is known for the latter 2 species in AD?
Answer: Thank you for this important question. Kocuria rhizophila is increasingly recognized as a commensal skin bacterium and has recently been detected in both healthy individuals and patients with atopic dermatitis. In paediatric AD cohorts, higher abundance of K. rhizophila has been associated with lower disease activity and improved barrier function, and in vitro studies suggest that K. rhizophila can inhibit Staphylococcus aureus growth. These findings support a potential protective or stabilizing role in the AD skin microbiome.
For human endogenous retrovirus K (HERV-K), much less is known in the context of AD. HERV-K is expressed in human skin and has been studied primarily in psoriasis and melanoma, where altered expression patterns have been reported. In AD, studies have mainly focused on other HERV families (e.g., HERV-E), and to our knowledge HERV-K expression has not been systematically characterized in AD skin. Our observations therefore add preliminary data to an area that remains largely unexplored. A couple of sentences about this have been included in the discussion section.
Figure 1A especially is rather busy. Can there be a summary of all 13 AD patients and may be also for Fig 1B for the control patients?
Answer: We appreciate the feedback regarding the figure appearing busy. However, we believe that this format remains the most effective way to present the relatively complex dataset while maintaining an accessible overview and preserving all relevant details for each sample.